# Speeding Up Speech Synthesis in Diffusion Models by Reducing Data Distribution Recovery Steps via Content Transfer.

## Abstract

Diffusion based vocoders have been criticised for being slow due to the many steps required during sampling. Moreover, the model's loss function that is popularly implemented is designed such that the target is the original input $x_0$ or error $\epsilon_0$. For early time steps of the reverse process, this results in large prediction errors, which can lead to speech distortions and increase the learning time. We propose a setup where the targets are the different outputs of forward process time steps with a goal to reduce the magnitude of prediction errors and reduce the training time. We use the different layers of a neural network (NN) to perform denoising by training them to learn to generate representations similar to the noised outputs in the forward process of the diffusion. The NN layers learn to progressively denoise the input in the reverse process until finally the final layer estimates the clean speech. To avoid 1:1 mapping between layers of the neural network and the forward process steps, we define a skip parameter $\tau > 1$ such that an NN layer is trained to cumulatively remove the noise injected in the $\tau$ steps in the forward process. This significantly reduces the number of data distribution recovery steps and, consequently, the time to generate speech. We show through extensive evaluation that the proposed technique generates high-fidelity speech in competitive time that outperforms current state-of-the-art tools. The proposed technique is also able to generalize well to unseen speech.

## 1 Introduction

The use of deep generative models is prevalent in speech synthesis (Lam et al., 2022) (Chen et al., 2020) (Kong et al., 2020b) (Prenger et al., 2018) (Kumar et al., 2019) (Kong et al., 2020a). These models use generative adversarial network (GAN) (Goodfellow, 2016) or likelihood-based techniques. GAN-based models such as (Kong et al., 2020a) and (Kumar et al., 2019) exploit the training objective to make the model generate data that are indistinguishable from the training data. While GAN based models can generate high quality speech, they are difficult to train due to instability during the training process (Mescheder et al., 2018). Likelihood speech synthesis-based techniques are composed of autoregressive models such as (Oord et al., 2016) (Kalchbrenner et al., 2018) (Mehri et al., 2016) (Valin & Skoglund, 2019), flow-based models (Prenger et al., 2019) (Kim et al., 2020) (Hsu & Lee, 2020) and variational auto-encoders (VAE) based models (Liu et al., 2022). Autoregressive speech synthesis models generate speech in a sequential nature, where the current sample to be generated is conditioned on the previously generated samples. Due to the sequential nature of speech generation, these models require many computations to generate a sample. This limits their ability to be deployed in application where faster real time generation is required. The flow-based model utilises specialised architectures to model a normalised probability model. These architectures require optimisation of many parameters during training, and hence can be computationally expensive. VAE based models on the other hand do not work well with high dimensional data (Bond-Taylor et al., 2021). Another type of likelihood-based generative model that is becoming popular for speech synthesis is the diffusion probability model (DPM) (Sohl-Dickstein et al., 2015). It has been explored in speech synthesis in (Lam et al., 2022) (Chen et al., 2020) (Kong et al., 2020b). DPMs are composed of two main processes i.e., the forward and

reverse process. The forward process involves sequentially adding Gaussian noise to a given distribution until eventually it becomes identical to white noise, i.e., pure Gaussian noise. The reverse process starts with white noise and recovers the data distribution by sampling. To learn a given target distribution, DPMs require a significant number of diffusion steps during training, resulting in many reverse steps to recover the data distribution during sampling time. Due to this, speech synthesis tools using diffusion generative model are slow, a property that prohibits their real-world deployment. Recognizing this limitation, speech synthesis tools that employ diffusion generative models employ a number of techniques to reduce sampling steps. WaveGrad (Chen et al., 2020) uses a grid search algorithm (GS) to reduce the sampling noise schedule. The use of grid search to shorten the noise schedule has been criticised for being computationally prohibitive when many noising steps $N$ are used (Lam et al., 2022). BDDM (Lam et al., 2022) reduces the sampling schedule by first training a generative model for speech synthesis using $T$ steps, then uses the optimised score network to train a scheduling network to learn a shorter noise schedule $N << T$ to be used during sampling. In this work, we explore the idea of using content transfer to speed up speech synthesis in DPM. Content transfer which was first used in (Gatys et al., 2016) as part of style transfer, involves training layers of a neural network to minimize the distance between representations of a desired style (or content) and a white noise and iteratively transform white noise to the desired style or content. Motivated by this, we also use neural network layers to learn to generate representations of a given audio generated by a given time-step $t$ of the forward process. Since the forward process can have many steps $T$, we restrict the layers of the neural network used in the reverse process to $N = \frac{T}{\tau}$ where $\tau > 1$. Intuitively, we use the layers of the neural network to reduce the noise schedule of the forward process by training a neural network such that its single layer can remove cumulative noise injected in $\tau$ steps during the forward process. Unlike (Lam et al., 2022) which optimises two sets of parameters, we train the model to optimise only a single parameter set $\theta$ therefore hypothesise that the proposed method will significantly reduce sampling time and, consequently, audio generation time.

## 1.1 Denoising diffusion probabilistic model

Given an observed sample $x$ of unknown distribution, the diffusion probabilistic model (DPM) aims to model the true distribution of the data $p(x)$. The modelled distribution $p(x)$ can then be used to generate new samples at will. DPM defines a forward process as:

$$q(x_{1:T}|x_0) = \prod_{i=1}^{T} q(x_t|x_{t-1}) \tag{1}$$

Here, latent variables and true data are represented as $x_t$ with $t = 0$ being the true data. The encoder $q(x_t|x_{t-1})$ seeks to convert the data distribution into a simple tractable distribution after the $T$ diffusion steps. $q(x_t|x_{t-1})$ models the hidden variables $x_t$ as linear Gaussian models with mean and standard centered around its previous hierarchical latent $x_{t-1}$. The mean and standard deviation can be modelled as hyperparameters (Ho et al., 2020) or as learnt variables (Nichol & Dhariwal, 2021) (Kingma et al., 2021). The Gaussian encoder's mean and variance are parameterized as $u_t(x_t) = \sqrt{\alpha_t}x_{t-1}$ and $\Sigma_q(x_t) = (1 - \alpha_t)I$ respectively, hence the encoder can be expressed as:

$$q(x_t \mid x_{t-1}) = \mathcal{N}(x_t; \sqrt{\alpha_t}x_{t-1}, (1 - \alpha_t)I) \tag{2}$$

$\alpha_t$ evolves with time $t$ based on a fixed or learnable schedule such that the final distribution $p(x_T)$ is a standard Gaussian. The encoder essentially describes a steady noisification of an input over time by adding Gaussian noise until eventually it becomes identical to pure noise. It is completely modelled as a Gaussian with a defined mean and variance parameters at each timestep hence it is not learned. Using the property of isotropic Gaussians, Ho et al. (2020) shows that $x_t$ can be derived directly on $x_0$ as:

$$x_t = \sqrt{\bar{\alpha}_t}x_0 + \sqrt{(1 - \bar{\alpha}_t)}\epsilon_0 \tag{3}$$

Where

$$\bar{\alpha}_t = \prod_{t=1}^{t} \alpha_t \text{ and } \epsilon_0 \sim \mathcal{N}(\epsilon_0; 0, I)$$

hence:

$$q(x_t|x_0) = \mathcal{N}(x_t; \sqrt{\bar{\alpha}_t}x_0, (1 - \bar{\alpha}_t)I) \tag{4}$$

The reverse process which seeks to recover the data distribution from the white noise $p(x_T)$ is modelled as:

$$p_\theta(x_{0:T}) = p(x_T) \prod_{i=1}^{T} p_\theta(x_{t-1}|x_t) \tag{5}$$

where

$$p(x_T) = \mathcal{N}(x_T; 0, I) \tag{6}$$

The goal of DPM is therefore to model the reverse process $p_\theta(x_{t-1}|x_t)$ so that it can be exploited to generate new data samples. After the DPM has been optimized, a sampling procedure entails sampling Gaussian noise from $p(x_T)$ and iteratively running the denoising transitions $p_\theta(x_{t-1}|x_t)$ for $T$ steps to generate $x_0$. To optimize DPM, evidence lower bound (ELBO) in equation 7 is used.

$$\log p(x) = E_{q(x_0)}[D_{KL}(q(x_T|x_0)||p(x_T)) + \sum_{t=2}^{T} E_{q(x_t|x_0)}[D_{KL}(q(x_{t-1}|x_t, x_0)||p_\theta(x_{t-1}|x_t))]$$
$$- E_{q(x_1|x_0)}[\log p_\theta(x_0|x_1)] \tag{7}$$

In equation 7, the second term on the right is the denoising term that seeks to model $p_\theta(x_{t-1}|x_t)$ to match the ground truth $q(x_{t-1}|x_t, x_0)$. In (Ho et al., 2020), $q(x_{t-1}|x_t, x_0)$ is derived as:

$$q(x_{t-1}|x_t, x_0)$$
$$= \mathcal{N}(\frac{\sqrt{\alpha}(1 - \bar{\alpha}_{t-1})x_t + \sqrt{\bar{\alpha}_{t-1}}(1 - \alpha_t)x_0}{(1 - \bar{\alpha}_t)}, \frac{(1 - \alpha_t)(1 - \bar{\alpha}_{t-1})}{(1 - \bar{\alpha}_t)}I) \tag{8}$$

In order to match $p_\theta(x_{t-1}|x_t)$ to $q(x_{t-1}|x_t, x_0)$ during the reverse process, $p_\theta(x_{t-1}|x_t)$ is modeled with the same variance as that of $q(x_{t-1}|x_t, x_0)$ i.e $\Sigma_q(t) = \frac{(1-\alpha_t)(1-\bar{\alpha}_{t-1})}{(1-\bar{\alpha}_t)}I$. The mean of $p_\theta(x_{t-1}|x_t)$ is made to match that of $q(x_{t-1}|x_t, x_0)$ hence it is parameterized as:

$$u_\theta(x_t, t) = \frac{\sqrt{\alpha}(1 - \bar{\alpha}_{t-1})x_t + \sqrt{\bar{\alpha}_{t-1}}(1 - \alpha_t)\hat{x}_\theta(x_t, t)}{(1 - \bar{\alpha}_t)} \tag{9}$$

Here, the score network $\hat{x}_\theta(x_t, t)$ is parameterized by a neural network and it seeks to predict $x_0$ from a noisy input $x_t$ and time index $t$. Hence,

$$p_\theta(x_{t-1}|x_t)$$
$$= \mathcal{N}(\frac{\sqrt{\alpha}(1 - \bar{\alpha}_{t-1})x_t + \sqrt{\bar{\alpha}_{t-1}}(1 - \alpha_t)\hat{x}_\theta(x_t, t)}{(1 - \bar{\alpha}_t)}, \frac{(1 - \alpha_t)(1 - \bar{\alpha}_{t-1})}{(1 - \bar{\alpha}_t)}I) \tag{10}$$

Therefore, optimizing the KL divergence between the two Gaussian distributions of $q(x_{t-1}|x_t, x_0)$ and $p_\theta(x_{t-1}|x_t)$ can be formulated as:

$$L_{t-1} = \arg\min_\theta E_{t \sim U(2,T)} D_{KL}(q(x_{t-1}|x_t, x_0)||p_\theta(x_{t-1}|x_t)) \tag{11}$$

$$L_{t-1} = \arg\min_\theta E_{t \sim U(2,T)} D_{KL}(\mathcal{N}(x_{t-1}; \mu_q, \Sigma_q(t)||\mathcal{N}(x_{t-1}; \mu_\theta(x_t, t), \Sigma_q(t)) \tag{12}$$

Here,

$$\mu_q = \frac{\sqrt{\alpha}(1 - \bar{\alpha}_{t-1})x_t + \sqrt{\bar{\alpha}_{t-1}}(1 - \alpha_t)x_0}{(1 - \bar{\alpha}_t)}$$

Equation 12 is simplified as (see (Luo, 2022)):

$$L_{t-1} = \arg\min_\theta E_{t \sim U(2,T)}[||\hat{x}_\theta(x_t, t) - x_0||_2^2] \tag{13}$$

Replacing $x_t$ in equation 13 as defined in equation 3 we have:

$$L_{t-1} = \arg\min_{\theta} E_{t \sim U(2,T)}[||\hat{x}_{\theta}(\sqrt{\bar{\alpha}_t}x_0 + \sqrt{(1 - \bar{\alpha}_t)}\epsilon_0, t) - x_0||_2^2] \tag{14}$$

The loss function is composed of the neural network $\hat{x}_{\theta}(x_t, t)$ that is conditioned on the discrete time $t$ and noisy input $x_t$ to predict the original ground truth input $x_0$. By rearranging equation 3 as:

$$x_0 = \frac{x_t - \sqrt{1 - \bar{\alpha}_t}\epsilon_0}{\sqrt{\bar{\alpha}_t}} \tag{15}$$

an equivalent optimization of modelling a neural network $\hat{\epsilon}_{\theta}(x_t, t)$ to predict the source noise can be derived (Ho et al., 2020).

$$L_{t-1} = \arg\min_{\theta} E_{t \sim U(2,T)}[||\hat{\epsilon}_{0\theta}(\sqrt{\bar{\alpha}_t}x_0 + \sqrt{(1 - \bar{\alpha}_t)}\epsilon_0, t) - \epsilon_0||_2^2] \tag{16}$$

Work in (Ho et al., 2020) uses $L_{t-1}$ as an optimization of the ELBO.

## 2 Related work

Deep neural network generative techniques for speech synthesis (vocoders) are either implemented using likelihood technique or generative adversarial network (Goodfellow, 2016). Likelihood methods are composed of autoregressive, VAE, flow, and diffusion-based vocoders. Autoregressive models such as (Oord et al., 2016) (Kalchbrenner et al., 2018) (Mehri et al., 2016) and (Valin & Skoglund, 2019) are models that generate speech sequentially. The models learn the joint probability over speech data by factorizing the distribution into a product of conditional probabilities over each sample. Due to their sequential nature of speech generation, autoregressive models require a large number of computations to generate a sample. This limits their ability to be deployed in application where faster real time generation is required. However, there are models such as (Paine et al., 2016), (Hsu & Lee, 2020) and (Mehri et al., 2016) which propose techniques to speed up speech generation in autoregressive models. Another likelihood-based speech synthesis technique is the flow-based models (Rezende & Mohamed, 2015) used in (Prenger et al., 2019) (Kim et al., 2020) (Hsu & Lee, 2020). These models use a sequence of invertible mappings to transform a given probability density. During sampling, flow-based models generate data from a probability distribution through the inverse of these transforms. Flow based models implement specialized models that are is complicated to train Tan et al. (2021). Denoising diffusion probabilistic models (DDPM) have recently been exploited in speech synthesis using tools such as PriorGrad (Lee et al., 2021), WaveGrad (Chen et al., 2020), BDDM (Lam et al., 2022) and DiffWave (Kong et al., 2020b). These models exploit a neural network that learns to predict the source noise that was used in the noisification process during the forward process. Diffusion-based vocoders can generate speech with very high voice quality but are slow due to the high number of sampling steps. Tools such as BDDM (Lam et al., 2022) propose techniques to speed up speech generation while using diffusion models. Our proposed work also looks at how to speed up speech synthesis in diffusion models. Finally, GAN based models such as (Kong et al., 2020a) and (Kumar et al., 2019) exploit the training objective to make the model generate data that is indistinguishable from the training data. While GAN based models can generate high quality speech, they are difficult to train due to instability during the training process (Mescheder et al., 2018). A complete review of the vocoders can be found in (Tan et al., 2021).

## 3 Unconditional speech generation

### 3.1 Forward Process

The proposed method is based on content transfer between audio in the forward process and audio generated by the layers of neural network in the reverse process. The number of steps $N$ used to recover the data distribution is fixed by selecting a skip parameter $1 \leq \tau < T$ such that $N = \frac{T}{\tau}$. If $\tau = 1$, then there is 1:1 mapping between a time step $t \in T$ of the forward process and a neural network layer of the reverse process. However, the goal is to fast track the reverse of the diffusion process, and hence we aim to select $\tau > 1$ that significantly reduces the sampling time. By doing this, a layer $l$ that is mapped to a time step $t$

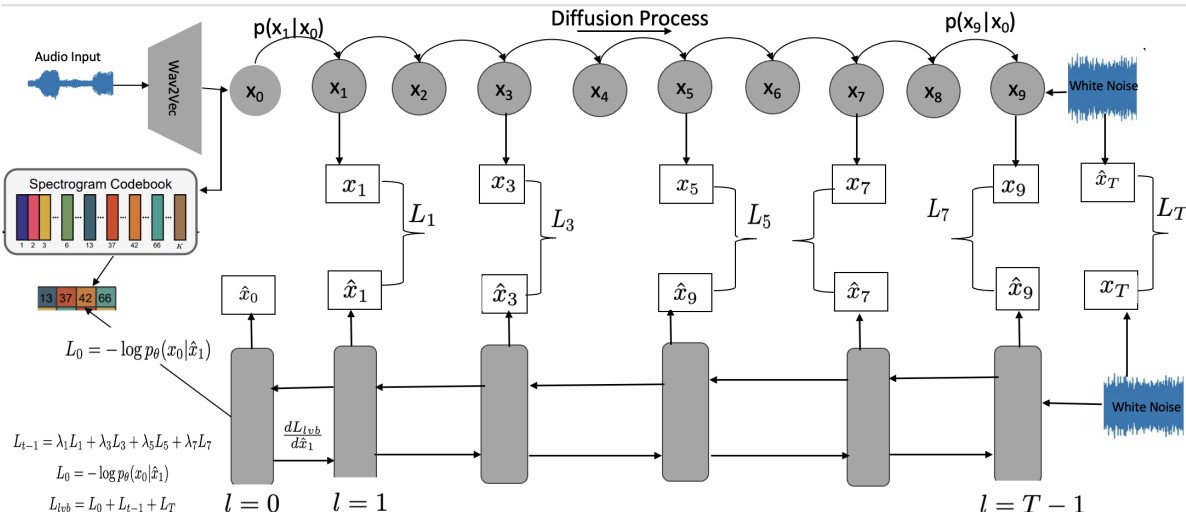

Figure 1: An overview of the unconditioned audio generation. An input audio is processed by a pretrained model to generate $x_0$. $x_0$ is then processed by forward process to generate latent variable $x_t$. The representations of latent variable $x_t$ are stored for selected steps. In the reverse process, white noise $x_T$ is passed through the first layer of the neural network and processed through the subsequent layers. For each layer, we store its generated representation. A layer is mapped to a given time step $t$ of the forward process. If a layer $l$ is mapped to a time step $t$, an error $L_i$ is computed by establishing $l_2$ norm between their respective embeddings.

of the forward process can eliminate noise injected from $t = t - \tau$ to $t = t$ in the diffusion process. During the forward process, given a raw waveform $x \in \mathbb{R}^T$, we use Wav2Vec 2.0 (Baevski et al., 2020) to establish its representations $x_0 \in \mathbb{R}^{f \times h}$. We then generate latent variables $x_t$ from $x_0$ based on equation 3. The representations of latent variables at selected steps $t$ are stored based on the set value of $\tau$. We also employ a discrete codebook $\mathcal{Z} = \{z_k\}_{k=1}^K \in \mathbb{R}^h$ such that the rows of $x_0$ are mapped to entries in the codebook based on equation 17 (see figure 1).

$$z_q = \left( \arg\min_{z_k \in \mathcal{Z}} ||x_0^i - z_k||_2^2 \text{ for all } (i) \text{in } (f) \right) \in \mathbb{R}^{f \times h} \tag{17}$$

The indexes from the codebook to which the row of $x_0$ are mapped to are stored.

## 3.2 Reverse process

The ELBO (equation 7) used for optimizing diffusion probabilistic model is composed of three key parts:

$$L_0 = E_{q(x_1|x_0)}[\log p_\theta(x_0|x_1)]$$
$$L_T = E_{q(x_0)} D_{KL}(q(x_T|x_0)||p(x_T))$$
$$L_{t-1} = \sum_{t=2}^{T} E_{q(x_t|x_0)}[D_{KL}(q(x_{t-1}|x_t, x_0)||p_\theta(x_{t-1}|x_t))]$$

The total loss based on these three parts is defined as:

$$L_{vlb} = L_0 + L_1 + \cdots + L_{T-1} + L_T. \tag{18}$$

$L_{t-1}$ is the denoising term with the goal to learn a transition step $p_\theta(x_{t-1}|x_t)$ that estimates the tractable ground truth $q(x_{t-1}|x_t, x_0)$. When the KL divergence of the two denoising steps is minimised, denoising is

achieved. $q(x_{t-1}|x_t, x_0)$ can be expanded using Bayes theorem as:

$$q(x_{t-1}|x_t, x_0) = \frac{q(x_t|x_{t-1}, x_0)q(x_{t-1}|x_0)}{q(x_t|x_0)} \tag{19}$$

Using Markov property,

$$q(x_t|x_{t-1}, x_0) = q(x_t|x_{t-1}) \tag{20}$$

Based on equation 3, $q(x_t|x_{t-1})$ can be derived directly from input as $q(x_t|x_0)$. Therefore, equation 19 is reformulated as:

$$q(x_{t-1}|x_t, x_0) = \frac{q(x_t|x_0)q(x_{t-1}|x_0)}{q(x_t|x_0)} = q(x_{t-1}|x_0) = q(x_{t-1}|x_{t-2}) \tag{21}$$

Hence equation $L_{t-1}$ can now be rewritten as:

$$L_{t-1} = \sum_{t=2}^{T} E_{q(x_t|x_0)}[D_{KL}(q(x_{t-1}|x_{t-2})||p_\theta(x_{t-1}|x_t))] \tag{22}$$

We further parameterize it as:

$$L_{t-1} = \sum_{t=2}^{T} E_{q(x_t|x_0)}[D_{KL}(q(x_{t-1}|x_{t-2})||p_\theta(\hat{x}_{t-1}|\hat{x}_t))] \tag{23}$$

Where we introduce estimates $\hat{x}_{t-1}$ and $\hat{x}_t$ instead of actual $x_{t-1}$ and $x_t$ respectively. The denoising goal now is to model $p_\theta(\hat{x}_{t-1}|\hat{x}_t)$ to estimate $q(x_{t-1}|x_{t-2})$) established during the forward process. In (Ho et al., 2020), they randomly sample $t$ and use $E_{t,x_0,\epsilon_0}[L_{t-1}]$(see equation 16) to estimate $L_{vlb}$ (equation 18). The minimization of the loss proposed by (Ho et al., 2020) generates samples that do not achieve competitive log-likelihoods (Nichol & Dhariwal, 2021). Log-likelihood is a key metric in generative models and optimizing it drives generative models to capture all of the modes of the data distribution (Razavi et al., 2019). Inspired by this, we seek to efficiently optimize the whole of $L_{vlb}$. We start by minimizing $L_{t-1}$ across all noise levels over all timesteps. This can be approximated by minimizing the expectation optimized using stochastic samples over timesteps.

$$L_{t-1} = \arg\min_\theta E_{t \sim U(2,T)} D_{KL}(q(x_{t-1}|x_{t-2}))||p_\theta(\hat{x}_{t-1}|\hat{x}_t)) \tag{24}$$

$$L_{t-1} = \arg\min_\theta E_{t \sim U(2,T)} D_{KL}(\mathcal{N}(x_{t-1}; \mu_q(t), \Sigma_q(t))||\mathcal{N}(\hat{x}_{t-1}; \hat{\mu}_\theta, \Sigma_q(t))) \tag{25}$$

where $\mu_q(t) = \sqrt{\alpha_{t-1}}x_{t-2}$, and $\Sigma_q(t) = (1 - \alpha_{t-1})I$. $p_\theta(\hat{x}_{t-1}|\hat{x}_t)$ is supposed to be modelled to have a similar distribution to that of $q(x_{t-1} | x_{t-2})$ as much as possible (Ho et al., 2020). Hence, we model the distribution of $p_\theta(\hat{x}_{t-1}|\hat{x}_t)$ as a Gaussian with mean $\hat{\mu}_\theta = \sqrt{\alpha_{t-1}}\hat{x}_\theta(\hat{x}_{t-1}, t - 2)$ and variance $\Sigma_q(t)$. With $\hat{x}_\theta(\hat{x}_{t-1}, t - 2)$ being parameterized by a neural network that seeks to predict $x_{t-2}$ from the estimate $\hat{x}_{t-1}$ and time step $t - 2$. Based on this, equation 25 can be simplified as ( see appendix A for complete derivation):

$$L_{t-1} = \arg\min_\theta E_{t \sim U(2,T)} \frac{1}{2(1 - \alpha_{t-1})}[||\sqrt{\alpha_{t-1}}\hat{x}_\theta(\hat{x}_{t-1}, t - 2) - \sqrt{\alpha_{t-1}}x_{t-2}||_2^2] \tag{26}$$

$$L_{t-1} = \arg\min_\theta E_{t \sim U(2,T)} \frac{\sqrt{\alpha_{t-1}}}{2(1 - \alpha_{t-1})}[||\hat{x}_\theta(\hat{x}_{t-1}, t - 2) - x_{t-2}||_2^2] \tag{27}$$

Equation 27 can be generalized as:

$$L_{t-1} = \arg\min_\theta E_{t \sim U(2,T)} \frac{\sqrt{\alpha_{t-1}}}{2(1 - \alpha_{t-1})}[||\hat{x}_\theta(\hat{x}_{t+1}, t) - x_t||_2^2] \tag{28}$$

Therefore, optimizing $L_{t-1}$ boils down to learning a neural network $\hat{x}_\theta$ to predict $x_t$ established during the forward process. The neural network should be conditioned on an estimate $\hat{x}_{t+1}$ and time step $t$ to predict $x_t$. Unlike the loss in equation 13 where the neural network $\hat{x}_\theta(x_t, t)$ conditioned on a random noisy input $x_t$

predicts the original noiseless input $x_0$, the loss in equation 28, conditions the neural network on an estimate $\hat{x}_{t+1}$ of step $t+1$ of the reverse process to predict the noisy output generated at a time step $t$ during the forward process. For instance, to recover $q(x_1|x_0)$ generated at $t = 1$ during the forward process, using the loss function in equation 28, we will need to design a neural network conditioned on an estimate $\hat{x}_2$ and time $t = 1$ and minimise the loss:

$$L_{t-1} = \arg\min_\theta E_{t\sim U(2,T)} \frac{\sqrt{\alpha_{t-1}}}{2(1-\alpha_{t-1})} [||\hat{x}_\theta(\hat{x}_2, 1) - x_1||_2^2] \tag{29}$$

### 3.3   Neural network

To recover data distribution, we use individual layers of Wav2Vec 2.0 (Baevski et al., 2020) to model a given timestep of the reverse process. Wav2Vec 2.0 is a speech pre-trained model that is composed of two main blocks. The first block, feature extractor is made of seven 1D convolution layers and a normalization layer. It accepts raw audio waveform and generates a representation $Z = \{z_1, \cdots, z_n\}$ with 20 ms stride between samples where each sample has a receptive field of 25ms. The second block is composed of a transformer with 24 layers that establish a contextual representation $C = \{c_1, \cdots, c_n\}$ of a given audio. Therefore, to recover data distribution, we use $N = 24$ steps where each step is modelled by a Wav2Vec's layer. This is similar to defining a skip parameter $\tau$ such that for a forward process with $T$, $N = \frac{T}{\tau}$. Factoring $\tau$ in $L_{t-1}$, equation 28 is implemented as:

$$L_{t-1} = \arg\min_\theta E_{t\sim U(2,T)} \frac{\sqrt{\alpha_{t-1}}}{2(1-\alpha_{t-1})} [||\hat{x}_\theta(\hat{x}_{t+1}, t) - x_{t-\tau}||_2^2] \tag{30}$$

Intuitively, since $\tau > 1$, the network layer $\hat{x}_\theta(\hat{x}_{n+1}, t)$ is supposed to remove the cumulative noise injected between timesteps $t = t - \tau$ and $t = t$ during the forward process. This significantly speeds the data recovery process. Note that since the layers of neural network are sequential, an estimate $\hat{x}_t$ generated by layer $l = t$ acts as input of layer $l = t - 1$ to estimate $\hat{x}_{t-1}$. Therefore, the timesteps $t$ are implicitly encoded by the layers hence the layers need not to be conditioned on time $t$ i.e.,

$$\arg\min_\theta E_{t\sim U(2,T)} \frac{\sqrt{\alpha_{t-1}}}{2(1-\alpha_{t-1})} [||\hat{x}_\theta(\hat{x}_{t+1}, t) - x_t||_2^2] = \arg\min_\theta E_{t\sim U(1,T-1)} \frac{\sqrt{\alpha_{t-1}}}{2(1-\alpha_{t-1})} ||\hat{x}_\theta^t(\hat{x}_{t+1}) - x_{t-\tau}||_2^2 \tag{31}$$

The reverse process starts with white noise $x_T \sim \mathcal{N}(0, I)$ which is passed through the first layer of neural network $l = T-1$ and subsequent layers of the Wav2Vec. Each layer $l = t$ generates representation estimation $\hat{x}_t \in R^{f\times h}$ which is passed to the next layer $l = t - 1$ to estimate $\hat{x}_{t-1} \in R^{f\times h}$. Representations encoded by each of the 24 layers of Wav2Vec are stored. We compute $L_{t-1}$ loss by minimizing $l2$ normalized embeddings between $\hat{x}_t \in R^{f\times h}$ and the embeddings $x_{t-\tau} \in R^{f\times h}$ established during the forward process according to equation 32.

$$L_{t-1} = \sum_{l=T-1}^{1} \lambda_i \sum_{i=1}^{f} 2 - 2 \frac{<\hat{x}_{ti}, x_{(t-\tau)i}>}{||\hat{x}_{ti}||_2.||x_{(t-\tau)i}||_2} \tag{32}$$

Here, $\lambda_t$ is the weight of the amount of loss contributed by a layer $l = t$ to the loss $L_{t-1}$. During implementation we drop $\frac{\sqrt{\alpha_{t-1}}}{2(1-\alpha_{t-1})}$ since it does not contribute significantly to the loss.

To compute the loss $L_0$, we pass the estimate $\hat{x}_1$ estimated by layer $l = 1$ to the last layer $l = 0$ of the neural network i.e $\hat{x}_0 = \hat{x}_\theta^0(\hat{x}_1)$. It is this predicted $\hat{x}_0$ that used to estimate the probability $p_\theta(\hat{x}_0|\hat{x}_1)$ of predicting the original indexes of the input $x_0$ established by the codebook (see figure 1). $L_0$ is then computed as:

$$L_0 = -\log p_\theta(\hat{x}_0|\hat{x}_1) \tag{33}$$

$L_T$ is not modelled by a neural network hence does not depend on $\theta$, it will be close to zero if the forward noising process sufficiently corrupts the data distribution so that $q(x_T|x_0) \approx \mathcal{N}(0, I)$. $L_T$ can simply be computed as KL Divergence between two Gaussian distributions. Based on this, we can now compute $L_{lvb} = L_0 + L_{t-1} + L_T$. The weights of the layers are by adjusted by computing the derivative of $L_{lvb}$ with respect to their activations.

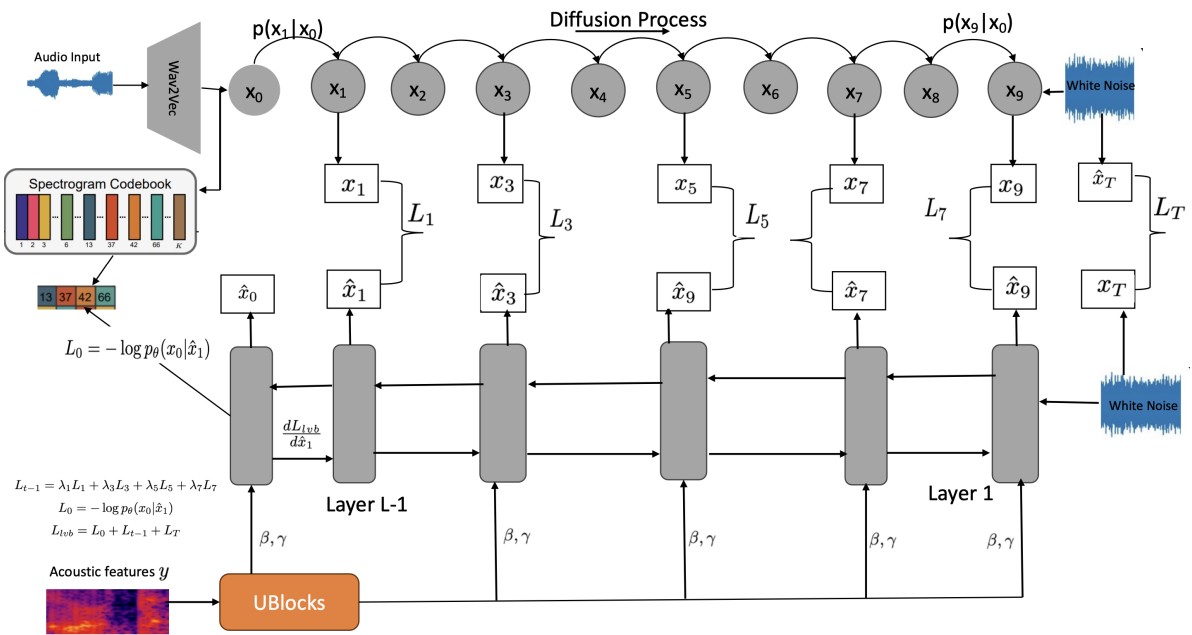

Figure 2: An overview of the conditioned audio generation. Compared to unconditioned audio generation in figure 1, the conditioned audio generation includes upsampling blocks that accepts Mel-spectrogram $y$ that contains acoustic features of the audio to be generated. The upsampling blocks process the Mel-spectrogram to generate $\beta = f(y)$ and $\gamma = f(y)$. Both $\beta$ and $\gamma$ are used to modulate the activation of the neural network layer according to equation 35.

### 3.4 Conditional speech generation

To enable the model to generate speech that follows a given acoustic features, we condition the loss in equation 31 on acoustic features $y$ as:

$$L_{t-1} = \arg\min_{\theta} E_{t \sim U(2,T)} \frac{\sqrt{\alpha_{t-1}}}{2(1-\alpha_{t-1})} ||\hat{x}_\theta^t(\hat{x}_{t+1}, y) - x_{t-\tau}||_2^2 \tag{34}$$

Therefore, we design the score network $\hat{x}_\theta^t(.,.)$ such that it can process both noisy estimate $\hat{x}_{n+1}$ and acoustic features $y$. To achieve this, we exploit feature-wise linear modulation (FiLM) (Perez et al., 2018) which has also been used in (Chen et al., 2020). Through FiLM, we adaptively influence neural network layer estimates by applying affine transformation to layer activation based on the input Mel spectrogram $y$ (see equation 35 and Figure 2). To compute $L_0$ for conditional generation, we estimate the probability $p_\theta(\hat{x}_0|\hat{x}_1, y)$ of predicting the original indexes of the input $x_0$ established by the codebook. Here, $\hat{x}_0$ is conditioned on both $\hat{x}_1$ and acoustic features $y$.

$$FiLM(\hat{x}_{n+1}) = \gamma \odot \hat{x}_{n+1} + \beta \tag{35}$$

where both $\gamma$ and $\beta \in R^{f \times h}$ modulates $\hat{x}_{n+1}$ based on a certain Mel-spectrogram $y$ and $\odot$ is the Hadamard product.

## 4 Evaluation

This section discusses how we developed and evaluated the proposed technique which we refer as **DiCon** (**Di**noising by **Con**tent transfer).

**Dataset**: In keeping with the trend in the speech synthesis domain and to allow for comparison with other existing tools, we used the most popular datasets i.e., LJSpeech dataset for single speaker evaluation and VCTK dataset for multi-speaker evaluation. LJSpeech dataset consists of 13,100 audio clips sampled at 22KHz. The audio clips are from a female speaker that vary in length from 1 to 10 seconds with a total

of about 24hrs. For multispeaker, we used the VCTK dataset, which is sampled at 48KHz and consists of 109 English speakers with various accents. VCTK was downsampled to 22KHz. Similar to (Chen et al., 2020), for LJSpeech, we used 12,764 utterance subset which is approximately 23 hours for training the model and evaluated it on test set of 130 utterances subset. For multi-speaker evaluation, we used the data split used in (Lam et al., 2022) of 100 speakers for training and 9 were used for evaluation. From each audio we extracted a 128-dimensional Mel spectrogram features. Similar to (Chen et al., 2020) we used a 50-ms Hanning window, 12.5-ms frame shift, and a 2048-point FFT with upper and lower frequencies of 20 Hz and 12 kHz lower.

**Training parameters**: The model was trained using a single NVDIA V100 GPU. We used Adam Optimiser and the cyclical learning rate (Smith, 2017) with a minimum learning rate of $1e-4$ and a maximum of $1e-1$. We used a batch size of 32 and trained for 1M steps. Similar to (Chen et al., 2020), for conditioned audio generation we used Mel-spectrogram extracted from ground truth audio as conditioning audio features during training while during testing we used Mel-spectrogram generated by Tacotron 2 model (Shen et al., 2018). To generate the FiLM parameters $\beta$ and $\gamma$, we use the upsampling blocks proposed in Chen et al. (2020) and use the parameters to modulate the activations of a given layer as described in equation 30. To weigh the loss contributed by a layer $l$ to the loss $L_{t-1}$ as described in equation 32, we initialize $\lambda = 0.001$ for the first layer and increase it by 0.001 for each upper layers i.e., upper layers contribute more to the loss as compared to lower layers.

**Baseline models**: We compared the proposed method with other state-of-the-art vocoders. We used models that have publicly available samples which can be used for human evaluation. The baseline models used include WaveNet (Oord et al., 2016)[1], WaveGlow (Prenger et al., 2018) [2], MelGAN (Kumar et al., 2019) [3], HiFi-GAN (Kong et al., 2020a) [4], WaveGrad (Chen et al., 2020) [5], DiffWave (Kong et al., 2020b) [6], BDDM (Lam et al., 2022) [7] and FastDiff (Huang et al., 2022a) [8].

**Metrics**: For subjective evaluation, we used the Mean Opinion Score (MOS) metric to evaluate the performance of the proposed model compared to the baseline tools. For each model we collected samples generated by the model. We also randomly selected samples from original audio samples. Each of the samples was presented to human evaluators one at time for them to rate the quality of speech on its naturalness on a 5-point Mean Opinion Score (MOS) scale. The scores used were Bad: 1, Poor: 2, Fair: 3, Good: 4, Excellent: 5 with a rating increment of 0.5. A single evaluator was required to rate 10 samples. Human evaluators were contracted via Amazon Mechanical Turk where they were required to wear headphones and be English speakers. For objective evaluation, we use deep learning-based MOS prediction tools. We use SSL-MOS [9] (Cooper et al., 2022), MOSA-Net[10] (Zezario et al., 2022) and LDNet [11](Huang et al., 2022c). We use these tools since they are also adopted as baseline tools for MOS prediction in VoiceMOS challenge Huang et al. (2022b). SSL-MOS is a Wav2Vec based model that was finetuned for MOS prediction task by adding a linear layer on top of Wav2Vec model. MOSA-Net employs a cross-domain features such as T-F spectrogram, complex spectrogram, raw waveform and features extracted from SSL speech models to predict MOS of speech samples. LDNet is a mean opinion score (MOS) prediction tool that predicts the listener-wise perceived quality given the input speech. Speech samples submitted to the tool need to be annotated with listener identity. MOS scores are based on all listeners averages. We also use $F_0$ Frame Error (FFE) which measures the proportional of the generated speech whose pitch differs from the ground truth. To compute FFE, ground truth speech is required. We use Mel-spectrograms of ground truth speeches to generate FFE.

**Model Configurations**: To train the model, we experimented with different number of steps in the forwards process while the reverse steps were kept constant at 24. We experimented with forward step (fsteps) of 1200, 960, 720 and 240 while the reverse steps (rsteps) were kept constant at 24 hence we selected a skip parameter

---

[1]https://github.com/r9y9/wavenet_vocoder

[2]https:// github.com/NVIDIA/waveglow

[3]https://github.com/descriptinc/melgan-neurips

[4]https://github.com/jik876/hifi-gan

[5]https://github.com/tencent-ailab/bddm

[6]https://github.com/tencent-ailab/bddm

[7]https://github.com/tencent-ailab/bddm

[8]https://FastDiff.github.io/

[9]https://github.com/nii-yamagishilab/mos-finetune-ssl

[10]https://github.com/dhimasryan/MOSA-Net-Cross-Domain

[11]https://github.com/unilight/LDNet

$\tau = \{50, 40, 30, 10\}$ respectively. The model accepts a 0.3 second input of audio. For the forward process $\alpha_i$ increases linearly from $\alpha_1$ to $\alpha_N$ defined as $Linear(\alpha_1, \alpha_N, N)$ such as $Linear(1 \times 10^{-4}, 0.005, 1200)$.

## 4.1 Results

### 4.1.1 Single speaker

For conditional speech generation on a single speech dataset, the subjective MOS and objective MOS from SSL-MOS, MOSA-Net and LDNet are shown table 1. Table 1 also shows the speech generation time RTF. For subjective MOS, the best performing configuration of the proposed technique DiCon(1200,24) registers a score 4.49 which has margin of 0.23 from that of the ground truth. DiCon(1200,24) also registers the best results in all the objective MOS. With regards to speed of speech generation, all the configuration of the proposed model are competitive with the observation that as the step size $\tau$ becomes smaller the speed of generation increases. We hypothesize that this is because a neural network layer has a reduced load of the amount of noise it is supposed to remove. We also note that the more forward steps, the better quality of audio the model can generate.

Table 1: Evaluation results of the conditioned version of the proposed method and how it compares to other state of the art tools on the evaluation metrics when single-speaker dataset is used.

| Model | MOS(↑) | SSL-MOS(↑) | MOSANet(↑) | LDNet(↑) | FFE(↓) | RTF (↓) |
|---|---|---|---|---|---|---|
| **LJSpeech test-dataset** | | | | | | |
| Ground truth | 4.72±0.15 | 4.56 | 4.51 | 4.67 | - | |
| BDDM(12 steps) | 4.38±0.15 | 4.23 | 4.17 | 4.42 | 3.6% | 0.543 |
| DiffWave(200 steps) | 4.43± 0.13 | 4.31 | 4.28 | 4.36 | 2.6% | 5.9 |
| WaveGrad(1000 steps) | 4.32± 0.15 | 4.27 | 4.23 | 4.31 | 2.8% | 38.2 |
| HIFI-GAN | 4.26± 0.14 | 4.19 | 4.13 | 4.27 | 3.3% | 0.0134 |
| MelGAN | 3.49± 0.12 | 3.33 | 3.27 | 3.42 | 6.7% | 0.00396 |
| WaveGlow | 3.17± 0.14 | 3.12 | 3.09 | 3.14 | 7.3% | 0.0198 |
| WaveNet | 3.61 ± 0.15 | 3.51 | 3.47 | 3.54 | 6.3% | 318.6 |
| DiCon(fsteps:1200 rsteps 24 ) | **4.49**± 0.12 | **4.43** | **4.35** | **4.44** | **2.3**% | 0.0042 |
| DiCon(fsteps:960 rsteps 24 ) | 4.33 ± 0.15 | 4.283 | 4.23 | 4.31 | 3.7% | 0.00371 |
| DiCon(fsteps:720 rsteps 24 ) | 4.17 ± 0.15 | 4.12 | 4.09 | 4.14 | 4.3% | 0.002912 |
| DiCon(fsteps:240 rsteps 24 ) | 4.09± 0.13 | 4.05 | 4.01 | 4.05 | 4.7% | **0.00182** |

### 4.1.2 Multi-speaker

The results of the performance of the proposed technique on the multi-speaker dataset are shown in table 2. For this dataset, the proposed technique can generalize to unseen speakers and DiCon(1200,24) configuration has the best MOS score of 4.38 which has a gap of 0.25 from the ground truth. It also registers the best performance on all the objective metrics except in FFE where it has a margin of 0.1% from the best performing tool on this metric DiffWave.

### 4.1.3 Unconditional speech generation

Here, the model was trained using multi-speaker dataset. To generate a speech sample, we sample white noise at random and process it through the trained model without conditioning it on any acoustic features. The results for unconditional speech generation are shown in table 3. For short clips, DiCon(fsteps:1200 rsteps 24 ) attains the MOS score of 3.11. Listening to the audio clips, we noticed a phenomenon where the clips begin by generating coherent sounding sentences, but the coherence drops with time. We will investigate the reason for this phenomenon in our future work. However, the model can generate clean sounding speeches, i.e., almost free of noise or artefacts.

Table 2: Evaluation results of the conditioned version of the proposed method and how it compares to other state of the art tools on the evaluation metrics when single-speaker dataset is used.

| VCTK test-dataset | | | | | | |
|---|---|---|---|---|---|---|
| **Model** | **MOS(↑)** | **SSL-MOS(↑)** | **MOSANet(↑)** | **LDNet(↑)** | **FFE(↓)** | **RTF (↓)** |
| Ground truth | 4.63±0.05 | 4.57 | 4.69 | 4.65 | - | |
| BDDM(12 steps) | 4.33±0.05 | 4.28 | 4.25 | 4.35 | 4.3% | 0.543 |
| DiffWave(200 steps) | **4.38**± 0.03 | 4.41 | 4.32 | 4.33 | **3.2**% | 5.9 |
| WaveGrad(1000 steps) | 4.26± 0.05 | 4.31 | 4.21 | 4.24 | 3.4% | 38.2 |
| HIFI-GAN | 4.19± 0.14 | 4.12 | 4.16 | 4.18 | 3.9% | 0.0134 |
| MelGAN | 3.33± 0.05 | 3.27 | 3.24 | 3.37 | 7.7% | 0.00396 |
| WaveGlow | 3.13± 0.05 | 3.12 | 3.16 | 3.09 | 8.2% | 0.0198 |
| WaveNet | 3.53 ± 0.05 | 3.43 | 3.45 | 3.46 | 7.2% | 318.6 |
| DiCon(fsteps:1200 rsteps 24 ) | **4.38**± 0.12 | **4.43** | **4.36** | **4.40** | 3.3% | 0.0042 |
| DiCon(fsteps:960 rsteps 24 ) | 4.28 ± 0.05 | 4.23 | 4.25 | 4.29 | 4.2% | 0.00371 |
| DiCon(fsteps:720 rsteps 24 ) | 4.12 ± 0.05 | 4.11 | 4.13 | 4.08 | 4.6% | 0.002912 |
| DiCon(fsteps:240 rsteps 24 ) | 4.04± 0.03 | 4.01 | 3.91 | 4.01 | 5.2% | **0.00182** |

Table 3: Results of the unconditioned proposed method on multi-speaker dataset.

| VCTK test-dataset | | | | | |
|---|---|---|---|---|---|
| **Model** | **MOS(↑)** | **SSL-MOS(↑)** | **MOSANet(↑)** | **LDNet(↑)** | **RTF (↓)** |
| DiCon(fsteps:1200 rsteps 24 ) | **3.11**± 0.12 | **3.17** | **3.16** | **3.23** | 0.0038 |
| DiCon(fsteps:960 rsteps 24 ) | 3.04± 0.05 | 3.09 | 3.01 | 3.09 | 0.00351 |
| DiCon(fsteps:720 rsteps 24 ) | 3.02 ± 0.05 | 3.07 | 3.01 | 3.06 | 0.002812 |
| DiCon(fsteps:240 rsteps 24 ) | 2.98± 0.03 | 3.02 | 3.03 | 3.08 | **0.00162** |

## 5  Conclusion

This paper presents DiCon, a technique for speeding up speech generation in diffusion models using neural network layers. We exploit the layers of the neural network to progressively recover the data distribution from white noise. Using content transfer, we demonstrate how an NN network layer can be exploited to implicitly perform denoising. We use a skip parameter $\tau$ to guide the mapping of NN layers to the forward process, and hence reduce the number of distribution recovery steps. In conditional speech generation, we use FiLM to infuse the acoustic features of a given speech into the denoising process. Based on evaluation, we demonstrate that the proposed technique generates superior quality speech samples at a competitive speed.

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

## 6 Appendix A

This section describes how equation 26 was arrived at.

$$L_{t-1} = \arg\min_{\theta} E_{t \sim U(2,T)} D_{KL}(q(x_{t-1} \mid x_{t-2}) || p_\theta(\hat{x}_{t-1}|\hat{x}_t)) \tag{36}$$

$$L_{t-1} = \arg\min_{\theta} E_{t \sim U(2,T)} D_{KL}(\mathcal{N}(x_{t-1}; \mu_q(t), \Sigma_q(t)) || \mathcal{N}(\hat{x}_{t-1}; \hat{\mu}_\theta, \Sigma_q(t))) \tag{37}$$

where $\Sigma_q(t) = 1 - \alpha_{t-1}$, $\hat{\mu}_\theta = \sqrt{\alpha_{t-1}} \hat{x}_\theta(\hat{x}_{t-1}, t-2)$ and $\mu_q(t) = \sqrt{\alpha_{t-1}} x_{t-2}$

$$D_{\mathrm{KL}}\left(\mathcal{N}\left(x_{t-1}; \mu_q, \Sigma_q(t)\right) \| \mathcal{N}\left(\hat{x}_{t-1}; \hat{\mu}_\theta, \Sigma_q(t)\right)\right) =$$
$$\frac{1}{2}\left[\log \frac{|\Sigma_q(t)|}{|\Sigma_q(t)|} - d + \mathrm{tr}\left(\Sigma_q(t)^{-1} \Sigma_q(t)\right) + (\hat{\mu}_\theta - \mu_q(t))^T \Sigma_q(t)^{-1} (\hat{\mu}_\theta - \mu_q(t))\right] \tag{38}$$

Here $d$ is the dimension of both distributions.

$$D_{\mathrm{KL}}\left(\mathcal{N}\left(x_{t-1}; \mu_q(t), \Sigma_q(t)\right) \| \mathcal{N}\left(\hat{x}_{t-1}; \hat{\mu}_\theta, \Sigma_q(t)\right)\right) = \frac{1}{2}\left[\log 1 - d + d + (\hat{\mu}_\theta - \mu_q(t))^T \Sigma_q(t)^{-1} (\hat{\mu}_\theta - \mu_q(t))\right] \tag{39}$$

$$= \frac{1}{2}\left[(\hat{\mu}_\theta - \mu_q(t))^T \Sigma_q(t)^{-1} (\hat{\mu}_\theta - \mu_q(t))\right] \tag{40}$$

$$= \frac{1}{2\Sigma_q(t)}\left[(\hat{\mu}_\theta - \mu_q(t))^T (\hat{\mu}_\theta - \mu_q(t))\right] \tag{41}$$

$$= \frac{1}{2\Sigma_q(t)}\left[\|\hat{\mu}_\theta - \mu_q(t)\|_2^2\right] \tag{42}$$

$$= \frac{1}{2\Sigma_q(t)}\left[\|\sqrt{\alpha_t}\hat{x}_\theta(\hat{x}_{t-1}, t-2) - \sqrt{\alpha_{t-1}} x_{t-2}\|_2^2\right] \tag{43}$$

$$= \frac{\sqrt{\alpha_{t-1}}}{2(1 - \alpha_{t-1})}\left[\|\hat{x}_\theta(\hat{x}_{t-1}, t-2) - x_{t-2}\|_2^2\right] \tag{44}$$

