# OpenReview forum: "Speeding Up Speech Synthesis in Diffusion Models by Reducing Data Distribution Recovery Steps via Content Transfer."
_TMLR — Rejected by TMLR_

### Review · Reviewer_RUzE · 2023-11-08

**Summary Of Contributions:**

This paper proposes a method for improving inference time/diffusion steps in diffusion-based neural vocoders. The authors propose to leverage the neural network's layers to predict several denoising steps at once. With that, the authors allow fewer reverse diffusion steps than forward steps. The authors compare the proposed method to several vocoders (diffusion and GAN-based ones), and show the superiority of the proposed approach when considering both generation quality and run time.

**Audience:**

Yes

**Broader Impact Concerns:**

The authors did not discuss the border impact of their method, which is probably involved with generating fake content.

**Claims And Evidence:**

No

**Requested Changes:**

1. Results on the multi-speaker datasets look strange. The model gets significantly better results when not trained on the multi-speaker dataset. This feels a bit strange and makes this method not very appealing. Moreover, in Table 3, the authors present results for the proposed method only. I assume this is because the baseline methods achieve much better results. Can the authors clarify that?

2. The authors present results for generalization to unseen speakers in Table 2 and compare the performance to several baseline methods. Although it is nice to see that all methods can produce good samples for unseen speakers, it is unclear what makes it special in the proposed method. From what I understand the main contribution of this submission is fast decoding, so it is not clear what should we learn from this set of results, especially since all methods perform roughly the same.

3. In many of the equations the authors use $E_q(x_n | x_{n−1})$. Shouldn't it be all in the sub-script? meaning: $E_{q(x_n | x_{n−1})}$? In case not, can the authors please clarify what they mean by this notation: $E_q(x_n | x_{n−1})[D_{KL}...]$?

4. The authors did not discuss the limitations of the proposed method. For instance: training time should be longer in their setup, as we need to pass over all diffusion steps during training time. During inference, we can not do more deciding steps to improve quality, as there are no more layers in the model. Meaning we are bounded by the network depth.

5. The authors state: “The model accepts a 0.3-second input of audio”. 0.3 sec seems relatively short segments (this is equivalent to roughly a phoneme), did the authors explore other/larger values than that?

6. The authors did not provide any speech samples. It is impossible to evaluate a speech generation paper without listening to the generated samples.

**Strengths And Weaknesses:**

**Strengths**:
* Making the inference of diffusion models faster is an interesting and important line of research, especially for the applicability of neural vocoders.
* The proposed method is simple to use.
* Results on single-speaker datasets look good. The authors include several evaluation metrics.
* The authors provide a detailed background on diffusion models, which makes this paper almost self-contained.

**Weaknesses**:
* It seems the model achieves poor performance on multi-speaker datasets.
* Some of the results are presented but it is not clear how they contribute to the overall contribution of this method.
* The paper does not discuss the limitations of the proposed approach.
* Some of the notations are unclear.

---

> ### Author Response · Authors · 2023-11-22
> **Rebuttal**
>
> -Results concern: We have re-evaluated the proposed tool on new metric ( suggested by reviewer SBKH) that are more relevant to speech synthesis.
>
> -We have also corrected all typos and errors as suggested.
>
> Comment: The authors did not discuss the limitations of the proposed method. For instance: training time should be longer in their setup, as we need to pass over all diffusion steps during training time.
>
> Response: Note that for forward steps q(x_t|x_0), no training is needed the latent variables x_t are generated directly from $x_0$ for all steps. This is fast.
>
> Comment: During inference, we can not do more deciding steps to improve quality, as there are no more layers in the model. Meaning we are bounded by the network depth.
>
> Response: This is accurate. We have captured this limitation  on the revision.

---

> > ### Comment · Reviewer_RUzE · 2024-01-07
> > **Response to authors**
> >
> > I would like to thank the authors for their response. The authors addressed most of my concerns. However, the authors still did not share any samples of the proposed method or compare them to the baseline methods.
> > As the authors mainly evaluate the proposed method using subjective tests (i.e., different MOS variations), I believe sharing samples is crucial to better understand the difference between the methods.
> >
> > I still have a few questions regarding this submission:
> > 1. Regarding the limitation of the decoding steps. I might have missed that, but I did not find a limitation section discussing this issue.
> > 2. Following my prev. question regarding unseen speakers, I highly encourage the authors to address/clarify this message. As I understand it, the main contribution of the proposed approach is speeding up the inference time of diffusion vocoders, hence it is not clear what is the purpose of the experiments over unseen speakers. It makes the impression that there is something specific in the proposed method allowing that.
> > 3. Regarding the unconditional generation experiment. It seems the authors do not condition the generation process on a spectral representation, in this case, I expect the output signal won't be intelligible and won't contain meaningful words. If that's true I expect the authors to discuss it in the manuscript, if not, can the authors additionally report WER.

---

### Review · Reviewer_SBKH · 2023-11-17

**Summary Of Contributions:**

This submission:

1) proposes a mechanism to speed up the reverse process of a class of diffusion models for spectrogram-to-waveform conversion

2) compares the resulting diffusion model to a wide range of other diffusion models for spectrogram-to-waveform conversion in a single, multi-speaker, and unconditional setting

**Audience:**

Yes

**Broader Impact Concerns:**

I do not believe there are any broader impact concerns related to this submission.

**Claims And Evidence:**

Yes

**Requested Changes:**

Fix all incorrect indices (e.g. eq 1)

Fix all incorrect typesetting of expectations (e.g. E_q(x_1|x_0) vs E_{q(x_1|x_0)} in eq 5).

Place citations where equations do not originate as the contribution of this submission (e.g. eq 5).

Make definitions of Gaussian distributions consistent (e.g. eq 7 vs eq 8).

Update section 3 to exclude elsewhere mentioned material and focus solely on the methods that aim to accomplish a similar task.

Merge section 4 into section 5.

Introduce at least some structure to otherwise completely unstructured writing in section 5.2.

Express equation 13 differently to avoid putting equal sign where other signs are perhaps more appropriate. Discuss the relationship between the left and the right hand side.

Explain clearly what kind of Markov property you are using to convert eq 14 into eq 15.

Explains clearly the motivation behind the transformation of equation 13. Explain clearly the role of \tau and why it does not feature anywhere other than the end of this section.

Remove L, = and argmin in equations 18 - 24 and focus on the function you are minimising. Clearly state whether you are minimising for one value of n at a time or you are minimising the average over n.

Change hadamard to Hadamard.

Clearly explain the design of NN layer which is capable of yielding both the embeddings contrasted to those generated by wav2vec as well as an output passed to the subsequent layer. Provide a clear description of the overall model.

Please note that your writing in section 5 is really hard to follow. A considerable improvement in quality is expected from any revision made to this submission.

**Strengths And Weaknesses:**

Strengths

1) This submission investigates an interesting constraint between forward and reverse processes which I believe have not been considered before.

Weaknesses

1) Technical description is poorly structured, contains numerous inconsistencies and is hard to comprehend.

2) Only waveform generation is considered making the impact limited to only those researchers working directly on this specific problem.

---

> ### Author Response · Authors · 2023-11-22
> **Rebuttal**
>
> R1: Thank you for your comments. We especially appreciate the comment on the paper lacking structure hence hard to understand. We have therefore made substantial effort introduce structure in the proposed technique. We  hope by the changes made the the paper will be more comprehensible.
>
> R2: We have corrected all suggested typos and equation errors.
>
> R3: We have still limited our contribution to speech generation. Even though the proposed technique can be easily be portable to image generation.

---

### Review · Reviewer_STUY · 2023-11-20

**Summary Of Contributions:**

This paper proposes a novel diffusion-based TTS system. Although diffusion-based methods can generate high-quality speech signals compared with the other methods, they require many iterations during inference. The proposed system can reduce the iterations and achieve both high-quality and fast inference using the popular LJSpeech and VCTK benchmarks.

**Audience:**

Yes

**Broader Impact Concerns:**

There is insufficient information on how Amazon Mechanical Turk is used (e.g., whether we need IRB approval since it involves human subjects, whether the wage is correctly set, etc.).

**Claims And Evidence:**

No

**Requested Changes:**

Major suggestions
- I recommend the authors rewrite the equation entirely, introducing all variables with explanations, making the formulation part self-consistent (without using the undefined variables used in the other papers), and making all variables consistent. I list some examples below, but simply fixing them is not sufficient. Please revise the formulation part of the paper entirely.
- Section 2: Please add more concrete examples in your vocoder application. The authors want to make Section 2 more general, but it is difficult to understand.
- In Eq. (10), $\epsilon$ suddenly appears without any definition.
- In Eq. (6), $\bar{\alpha}$ suddenly appears without any definition (It would be written in Ho et al. (2020), but then, this paper is not self-consistent).
- Section 5.1; What is $n$?
- Figure 1: some equations look ugly (the scales are not balanced). Please appropriately format them.
- Eq. (13): is this $n$ the same as that in Section 5.1?
- Eq. (13): I could not understand the first line. Why is the summation over $t$ performed to the variable that depends on $n$ (not $t$)?
- I could not find the explicit relationship between Eq. (5) and Eq. (13) as they look very different. The authors require more careful derivations about Eq. (13) from Eq. (5). Also, please justify the derivation. I could not understand why they exclude $x_0$.
- Eq. (15): I could not understand this derivation. Why can we introduce $q(x_n|x_{n-1})$ and $q(x_{n-1}|x_{n-2})$?
- Eq. (17): ditto. Why $t$ and $n$ are mixed?
- $R ^{k \times h}$: what is $h$?
- Section 6: Why did you use DNSMOS? It would be better to use MOSnet, or other DNN-based MOS developed for TTS quality evaluation (see the VoiceMOS challenge activities).
- Tables 1 and 2: the numbers are not aligned (with the period (.)). The number of significant figures is also not consistent.

Some detailed minor suggestions
- Please add more concrete numbers and databases to the abstract to claim the experimental results.
- Page 3: T steps --> $T$ steps.

**Strengths And Weaknesses:**

Strengths
- Diffusion-based TTS is a very hot topic
- The proposed method archives high-quality and fast inference using the popular LJSpeech and VCTK benchmarks compared with other vocoder methods (e.g., GAN, Flow, autoregressive, and other diffusion-based methods). The results are very impressive.

Weaknesses
- Frankly, I could not understand the main idea of the proposed method based on their inconsistent (or wrong) mathematical formulations. This is a very critical issue, and I could not validate their results due to this.

---

> ### Author Response · Authors · 2023-11-22
> **Rebuttal**
>
> Reviewer's Comment: Fankly, I could not understand the main idea of the proposed method based on their inconsistent (or wrong) mathematical formulations. This is a very critical issue, and I could not validate their results due to this.
>
> Response: We have made extensive effort to correct the writing of the paper with the goal to hopefully make it more flowing and clear.
> we have corrected all the errors proposed above
>
> Reviewer's comment:Why did you use DNSMOS? It would be better to use MOSnet, or other DNN-based MOS developed for TTS quality evaluation (see the VoiceMOS challenge activities).
>
> Response: Based on this comment, we have re-evaluated  the system on three MOS based DNN tools.

---

### Decision · Action_Editor_Dm7m · 2024-01-16

**Recommendation:** Reject

**Comment:**

All reviewers agree that the topic of the paper is of interest and the proposed approach, in which a neural network is trained to directly predict the outcome of multiple steps of the reverse diffusion process in an unrolled fashion, with successive layers in the network corresponding to equally spaced steps in the diffusion schedule, is creative and promising.

However, the reviewers also had substantial critiques of the original manuscript that are only partially addressed by the revision. Since the contents of the reviewers' official recommendations are not visible to the authors, I quote parts of them here because I believe that with more careful explication, this paper could be a good TMLR paper, and I want to assist the authors in improving the quality of the manuscript.

**Reviewer STUY:**

"Although the authors updated several derivation parts in the formulation, it still has ambiguous notations (e.g., the relationship between $t$, $h$, $n$, $l$, and $k$ are very confusing). These parts are critical to understanding their proposed methods but do not have sufficient improvements."

**Reviewer SBKH:**

"Despite making significant changes, the authors failed to address a large number of inconsistencies in their submission."

**Reviewer RUzE:**

"[T]he authors mainly evaluate the proposed method using subjective tests (i.e., different MOS variations), I believe sharing samples is crucial to better understand the difference between the methods."

The key messages here are (1) the importance of actually sharing audio samples, and (2) the need for more care in preparing the manuscript and ensuring that a reasonably knowledgable reader can understand the key points.

**Audience:**

The topic of accelerating inference in diffusion models is of broad interest to the TMLR audience, and the underlying approach suggested by the paper is a promising one, so the paper definitely meets this standard.

**Claims And Evidence:**

The claims made in the submission are not sufficiently supported. The paper falls short in two key aspects:
1. Despite thorough guidance from the reviewers in the first round of review, the paper's mathematical presentation is still not clear enough. Specifically, the notation is difficult to follow and the relationships between a number of key variables still must be made clearer.
2. The paper does not provide any audio samples, which is a bare minimum requirement for a work on speech synthesis.

**Resubmission Of Major Revision:**

The authors may consider submitting a major revision at a later time.